# Bird Beta Diversity in Sharp Contrasting Altai Landscapes: Locality Connectivity Is the Influential Factor on Community Composition

**DOI:** 10.3390/ani12182341

**Published:** 2022-09-08

**Authors:** Na Li, Yueqiang Liu, Hongjun Chu, Yingjie Qi, Xiaoge Ping, Chunwang Li, Yuehua Sun, Zhigang Jiang

**Affiliations:** 1Institute of Eastern-Himalaya Biodiversity Research, Dali University, Dali 671003, China; 2Key Laboratory of Animal Ecology and Conservation Biology, Institute of Zoology, Chinese Academy of Sciences, Beijing 100101, China; 3University of Chinese Academy of Sciences, Beijing 100049, China; 4Yunnan Appraisal Center for Ecological and Environmental Engineering, Kunming 650000, China; 5College of Ecology and Environment, Yunnan University, Kunming 650000, China; 6Institute of Forest Ecology, Xinjiang Academy of Forestry, 191 Anju South Road, Shuimogou District, Urumqi 830000, China; 7Xinjiang Uygur Autonomous Region Grassland Station, 618 Yanerwo Road, Tianshan District, Urumqi 830000, China

**Keywords:** bird, metacommunity, beta diversity, landscape, locality connectivity

## Abstract

**Simple Summary:**

Understanding the ecological processes involved in establishing a metacommunity, a collection of small communities linked by species dispersal, could help with biodiversity conservation. In this work, we examine how local community connectivity affects the distributional patterns of various bird species across metacommunities in the strikingly different Altai landscapes. We discovered that connectivity was most important for determining community composition in metacommunities. High beta diversity and a high turnover component in the research region showed that regional-scale conservation efforts should consider overall biodiversity. Although they are not part of the Altai-Sayan biodiversity ecoregion, the riparian and desert landscapes are essential to the birds that reside in the mountain ranges and should be regarded as integral parts of the ecoregion, and high connectivity stepping-stone habitats in these landscapes should be protected.

**Abstract:**

Located on the southwest slope and plain areas of the Altai Mountains in China, this study aims to explore bird composition variation (beta diversity) in mountain landscape (metacommunity M), riparian landscape (metacommunity R), desert landscape (metacommunity D) and across the three landscapes (metacommunity A), and to assess how patch connectivity with environmental and spatial factors influence species distributional patterns across multiple metacommunities. In 78 transect lines over the study area, 9724 detections of 139 bird species were detected. We calculated the beta diversity, its turnover and nestedness components in four metacommunities. We used the variation partitioning method to investigate the relative importance between the environment, spatial variation and locality connectivity in driving bird community composition variation. We found high beta diversities with a small contribution of nestedness components in all four metacommunities. When only a single set of predictors is contained in the model, the predictor that best explains the variation of bird community composition is connectivity in metacommunity M, R and D and spatial predictor in metacommunity A. In all three sets of predictors, 73.8~85.4% of variations of community composition can be explained in the four metacommunities, and connectivity always contributed the most. High beta diversity and a high turnover component imply that regional-scale conservation efforts should be thought of as preserving overall biodiversity. A conservation strategy is to keep stepping-stone habitats with good connectivity in the middle of the riparian landscape. Along with the Altai-Sayan biodiversity ecoregion, the desert and riparian environments are essential for birds residing in the mountainous terrain. Furthermore, they should be regarded as integral parts of the ecoregion.

## 1. Introduction

Ecological communities show complex patterns of variation in space. Understanding patterns of species composition and quantifying the relative importance of various factors in driving species composition variation is one of the main goals of community ecology [1] and is also an important aspect of regional biodiversity [2]. A metacommunity is defined as a set of local communities connected by dispersal of landscape or regional scale [3,4]. Since the metacommunity has connectivity between communities with different environments and locations by dispersal, it fuels research on environmental and spatial factors and the combined effects (environmental spatial structure, or space autocorrelation) on community diversity [5]. The basis of metacommunity theory and a key development in community ecology is that the makeup of a local community is influenced by both local and regional influences [6].

It is widely accepted that environmental, spatial and dispersal factors are mechanisms that drive beta diversity in a metacommunity [7]. The variation in species composition among sites, i.e., beta diversity [8,9], has been widely studied since 2001 [10]. Beta diversity comprises turnover and nestedness processes [11,12,13]. Turnover means another set replaces one set of species. Nestedness means one set of species is a subset of another set. The methodological framework that partitions beta diversity into turnover and nestedness components has value in measuring regional biodiversity with conservation implications [14].

However, few studies pay attention to dispersal dynamics (but see Ai et al., 2013; Monteiro et al., 2017) [7,15], while most quantitative approaches focus on the effect of local environmental factors or/and spatial variation (variation that is spatially autocorrelated) on the metacommunity variations (e.g., Cottenie 2005; Buchi et al., 2009; Henriques-Silva et al., 2013; Seymour et al., 2016; Tina et al., 2017) [1,5,16,17,18]. The spatial position of a locality (in this study, the term ‘locality’ is defined as an area of habitat encompassing multiple microsites and capable of holding a local community, which is assumed to be a random sample of species drawn from a metacommunity, following [19]) in the landscape, the environmental condition and species dispersal ability are found to influence the ability of a species to track its favorable environment. Recent studies find that species composition variation explained by space is usually due to dispersal dynamics [20]. At the same time, dispersal dynamics and missing spatialized environment factors are potentially confounded [21]. Multivariate models of community structure commonly explain a small fraction of composition variation, i.e., $R^{2}\;$< 50% and often between 0% and 20%. This may be because they disregard dispersal dynamics as well as unmeasured environmental influences [22]. Additionally, restoration and conservation biology may benefit from considering dispersal patterns. Ecological stepping-stones, or environments that aid in the dispersal of organisms from one place to another suitable site, could be one such example. To deal with global environmental changes, there are better ways to organize the conservation of metacommunities [22].

We aim to measure the impact of environmental factors, spatial location and dispersal dynamics on bird metacommunities in the Altai landscapes in northwest China. We predicted that bird species composition variated by the environment, spatial variation, locality connectivity (dispersal dynamics) or a combination of these factors. The study area has three landscapes: mountain, riparian and desert. We are keen to know whether birds in the three landscapes are isolated metacommunities or subsets of a larger metacommunity that includes birds living in the riparian areas, the desert and the mountains. We also investigated the factors that drive the diversity of bird metacommunities in the Altai landscapes by employing environmental predictors, spatial predictors, and locality connectivity as dispersal dynamics predictors. Examining the patterns of bird composition across numerous metacommunities in a wide geographical area could provide a way to test the existence of general principles underlying species distributions.

## 2. Methods

### 2.1. Study Area

The study area is located in the Altay region of China (85~91° E; 45~49° N). It is distinctly composed of the mountain, plain riparian and desert landscapes (Figure 1). This area has a temperate arid climate. The mountain landscape is located on the central and southwest slopes of the Altai Mountains, which is part of the Altai-Sayan biodiversity hot spot ecoregion (Conservation International, 2005). In these mountains, the mean annual temperature is −4~−2 °C, and the mean annual precipitation is 300~500 mm. The plain area contains riparian and desert landscapes, where the mean annual temperature is 4 °C and the mean annual precipitation is 100 mm. The natural soil types are mountainous chestnut soil, mountainous chernozem soil, mountainous gray forest soil and sub-high mountainous meadow soil on the mountains. The plains have cultivated meadow soil, bench land moist soil, meadow brown calcic soil and semifixed aeolian sandy soil. The original forests distribute along both sides of the river valleys in the mountain and plain riparian landscapes. The forests have a sparse natural vegetation coverage of around 30~40%.

The Altai Mountains present wide ranges of temperate forest-steppe zones, and the dominant trees are Siberian larch (*Larix sibirica*), Siberian spruce (*Picea obovata*), Siberian fir (*Abies sibirica*)*,* Eurasian aspen (*Populus tremula*), European white birch (*Betula pendula*) and laurel-leaf poplar (*Populus laurifolia*), and under the woody tress are juniper (*Juniperus* spp.), rose (*Rosa* spp.), honeysuckle (*Lonicera* spp.) and barberry (*Berberis* spp.). The plain area in front of the Altay Mountains is composed of two distinguished landscapes: riparian and desert landscapes. The riparian landscape is nourished by the Irtyshe River and Ulungur River, which originate from the Altai Mountains. The dominant species in the original riparian forest are salicaceous woods, including back poplar *(Populus nigra*), Erqisi poplar (*Populus jrtyschensis*), silver poplar (*Populus alba*), grey poplar (*Populus canescens*), laurel-leaf poplar (*Populus laurifolia*), Eurasian aspen (*Populus tremula*), desert poplar (*Populus euphratica*), and white willow (*Salix alba*); shrubs include almond willow (*Salix triandra*), grey willow (*Salix ciberea*), prickly wild rose (*Rosa acicularis*), common salt tree (*Halimodendron halodendron*), etc. The riparian landscape has been reclaimed as farmland and pastures. The vast desert landscape is located to the south of riparian landscape and covered by dwarf eremophyte, which mainly are saxaul (*Haloxylon ammodendron*), sagebrush (*Artemisia* spp.) and reed grass (*Calamagrostis* spp.), with sparse natural vegetation coverage of less than 10%. There is more snowfall than rainfall in the whole area, and the period of the snow season increases with elevation.

### 2.2. Bird Surveys

In this study, we set the transect lines at habitat patches with typical natural vegetation. In the mountains, the transect lines were set in seven valleys. In the riparian landscape, transect lines were set from upriver to downriver. In the desert landscape, transect lines were set from south to north and from west to east. The first transect in valleys, riparian and desert landscapes was randomly set; thereafter, a systematic scheme was followed with distance larger than 5 km between adjacent transect lines. Each transect line was 5 km long. The study aimed to set 25, 20 and 40 transect lines in mountain, riparian and desert landscape proportional to the areas. Finally, 78 transects were set, 22, 19 and 37 transect lines were set in mountain, riparian and desert landscapes, respectively, based on topographical structure. Bird surveys were conducted during the summer (June to August) in 2014, 2015 and 2016, respectively. Birds within 50 m on both sides of the transect lines were recorded because the visual range in woodlands is only about 50 m. With a 5 km transect, which is long enough to cover various local habitat types, we made up for the missing data of birds from 50 m distant. The researchers’ walking speed was controlled at 2 km/h. Each transect line was completed within 3 h, either in the morning after sunrise or in the afternoon before sunset on the days without wind or rain. The positions of all transect lines and bird-spotted points on each line were recorded by a global positioning system receiver (Garmin eTrex 30, China, Shanghai) with 5 m precision. Each line was visited twice, and the visiting time was reversed to avoid diurnal changes in bird activity. For each spot, bird species, numbers, distances to transect line and detect the position (latitude and longitude of the detect point) were recorded. The first author conducted bird counts, accompanied by two field assistants. Both the first author and field assistants were trained to identify local bird species in 2013 and 2014 before data collection.

### 2.3. Environmental and Spatial Variables Collection

For every bird-spotted point, the elevation and geographical factors of latitude and longitude were recorded with a GPS receiver in the field. The average values of the Enhanced Vegetation Index (EVI, http://www.gscloud.cn/, accessed on 25 December 2016) from 2010 to 2015 were used as a vegetation cover index. EVI is an optimized vegetation index and is responsive to canopy structure variations, such as leaf area index, canopy type, plant physiognomy and canopy architecture. The Compound Topographic Index (CTI) was downloaded from the United States Geological Survey’s Hydro 1K dataset network. CTI is a steady wetness index to quantify topographic control on the hydrological process, defined as $\ln\frac{a}{\mathrm{tanb}}$, where *a* is the local upslope area draining through a certain point per unit contour length and *tanb* is the local slope. The study used Human Footprint (HFP) as a human influence factor. HFP (http://sedac.ciesin.columbia.edu/wildareas/, accessed on 21 November 2016, Last of the Wild Data Version 2, 2005) is a quantitative analysis of human influence by overlaying population residence, land-use type, road network and infrastructure construction layers. 

Climate variables, including Annual Mean Temperature (AMT), Mean Diurnal Temperature Range (MDTR) and Annual Precipitation (AP), were downloaded from WorldClim (http://www.worldclim.org/, accessed on 15 November 2016). For each transect line (locality), the average values of environmental variables (including elevation, EVI, CTI, HFP and three climate variables mentioned above, 1 km resolution) of bird-spotted points within the transect line were used. Thus, for each bird locality, seven environmental variables were collected. The average values of latitude and longitude were used as spatial data.

### 2.4. Data Summary

The study summarized bird species, environmental and spatial data across localities in the mountain landscape (metacommunity M), the plain riparian landscape (metacommunity R), the desert landscape (metacommunity D) and localities across the three kinds of landscapes (metacommunity A). In each metacommunity, three metrics were built: a presence/absence locality×species matrix (rows are transect lines and columns are bird species), a locality×corrdinate matrix (columns are latitude and longitude) and a locality×environment matrix (columns are the seven environmental variables mentioned above). The locality×species matrix was used for calculating the beta diversity coefficients. The locality×coordinate matrix was used for calculating Moran’s Eigenvector Maps as spatial variables. The connectivity variables were calculated based on the locality×species matrix and locality×corrdinate matrix by the method following Monteiro et al. [15] (see below).

### 2.5. Beta Diversity

The study calculated bird beta diversity in metacommunities A, M, R, and D. Beta diversity values were measured by Sørensen’s dissimilarity index (beta.sor) and partitioned beta.sor into a turnover component (Simpson’s dissimilarity, beta.sim) and nestedness-resultant component (beta.nes) following Baselga [13]. Dissimilarity coefficients were calculated between every pair of localities within each metacommunity. The averages and standard errors were calculated to quantify between the metacommunity variability. Differences in average values were tested by using the Kruskal–Wallis and Mann–Whitney tests. The rate of beta.nes to beta.sor (beta.ratio) was used to measure the relative contribution of the spatial nestedness component to beta diversity.

### 2.6. Spatial, Environmental and Connectivity Predictors and Variation Partitioning

A variation partitioning framework for empirical metacommunity analysis is a method for distinguishing species sorting via differences in environmental affinities and dispersal (see Logue et al., 2011) [6]. The study used the variation partitioning framework improved by Monterio et al. [15] to distinguish the contributions of locality connectivity, environment and spatial variation to determine beta diversity in these four metacommunities. The analysis was carried out as follows: (1) The study calculated a nearest occupied site distance metric (metric N) for every bird species at every locality, and an average connectivity metric (metric C), which is used by Hanski and Singer [23]:$$c_{\mathrm{ik}}=\overset{n}{\underset{\begin{array}{c} i=1\\ i\neq j \end{array}}{\sum }}N_{\mathrm{jk}}\exp\left(-\frac{d_{\mathrm{ij}}}{\alpha_{k}}\right)$$
where c is the connectivity value for the kth species at the target locality i and every other locality j, N is the number of individuals in locality j for species k, d is the geographical distance between localities i and j, and $\alpha_{k}$ controls the steepness of the dispersal kernel in which small α values represent greater dispersal limitation in contrast to larger ones. A single α value was finally used across all species, which maximized the prediction of species distributions based on metric C and was estimated via iteration as suggested by Yamanaka et al. [24]. Metric N and C were used as connectivity predictors. (2) Moran’s Eigenvector Maps (db-MEM; [25]) calculated by geographical matrices were used as autocorrelation spatial predictors to estimate the spatial variability of community composition. The eigenvectors were also used in controlling for autocorrelation when testing the importance of environmental predictors. (3) For the environmental predictors, the study first tested the correlation between every pair of the seven variables. A strong correlation was found between EVI and AMT (Spearman test, r = −0.93), and the study removed AMT from the following analysis to reduce collinearity. Thus, six environmental variables were engaged. 

For each metacommunity, the study conducted the Redundancy Analysis (RDA; [26]) between the environmental matrix and species matrix to retain environmental variables that significantly affect avian species composition. The study used Mantel correlograms [27] to assess the spatial structure of the retained environmental variables. (4) Finally, the study used RDA to calculate the percentage of species variation in the locality—species matrix explained by environmental predictors (different environmental variables remained in four metacommunity, as shown in the Results section), spatial predictors (MEM) and connectivity predictors (metric N and metric C). The Monte Carlo permutest with 1000 random runs was used to examine the significance of the models. The connectivity of a certain locality was calculated as the sum of every species’ connectivity value in that locality. All of the analyses were performed in R 3.1.3 software using packages *vegan* and *betapart* (R Development Core Team, 2010).

## 3. Results

### 3.1. Bird Species Richness and Beta Diversity

A total of 9724 individuals of 139 bird species were recorded in the study areas (refer to Appendix A for species information), including 85 species in the mountains, 50 species in the desert and 97 species in the riparian landscape. The mountain landscape and riparian landscape shared 51 bird species, riparian and desert shared 35 species, while 29 bird species were detected in both mountain and desert, and 22 species were detected in all of the three landscapes (Figure 2). A total of 81 species were summer migrant birds. Bird species records in more than one transect line (107 of 139 species; Table 1) were used for beta diversity and variation partitioning analysis. All metacommunities had high beta diversities with small components of nestedness (Table 1). The highest beta diversity (beta.sor = 0.83) with the smallest component of nestedness (beta.ratio = 11.6%) was found in metacommunity A. Conversely, the lowest beta diversity (beta.sor = 0.58) with the highest nestedness component (beta.ratio = 28.7%) was found in the metacommunity R.

### 3.2. Spatial Structure of the Environment Variables

The environmental variables that affected bird species composition significantly (*p* < 0.05 by permutation test) were different among the four metacommunities (Appendix B). In metacommunity A, all of the six variables significantly affected bird species composition, and the EVI effects the most (R^2^ = 0.771, *p* = 0.001). In metacommunity M, the most effective environmental variables were MDTR (R^2^ = 0.716, *p* = 0.001), and other effective variables were HFP, CTI, elevation, and AP. In metacommunity R, the environmental variables that significantly affected bird species composition in the order from strong to weak were AP, MDTR elevation, and HFP. And in metacommunity D, effective environmental variables were elevation, MDTR, HFP, and AP.

Based on the effective environmental variables, the Mantel correlograms in the four metacommunities indicated that spatial environmental structures, localities with more similar environments were generally within the spatial distance of 100 km for metacommunity A, M and R, and 50 km for metacommunity D (Mantel r value > 0.2 and *p* < 0.05; Figure 3).

### 3.3. Contribution of Environment, Spatial Variation and Locality Connectivity to Bird Species Composition Variation

The contributions of environment, spatial and locality connectivity to bird species variation in metacommunities A, M, R and D were estimated. Three sets of predictors (separately) significantly explained the species composition variation in the study metacommunities. By one class of predictor in the model, the highest explained variation of bird composition was 21.9% in metacommunity A with spatial variation (*p* = 0.01), 34.9% in metacommunity M with connectivity (*p* = 0.01), 41.2% in metacommunity R by connectivity (*p* = 0.01) and 18.4% in metacommunity D with connectivity (*p* = 0.01). By all of the three predictors, the total percentage of explained variation was 73.8%, 83.5%, 85.4%, and 81.1% in metacommunities A, M, R and D, respectively (*p* = 0.01; Figure 4).

With all of the environmental, spatial and connectivity predictors in the model, variation partitioning was processed for metacommunities A, M, R and D. For all of the four metacommunities, the variation explained with connectivity was higher than those explained with environment and spatial predictors across all α values. Except in metacommunity A when α was smaller than 100 km, the spatial variation explained most. The variations explained by connectivity and all three types of predictors were both increased with the dispersal distance, and all were maximized at the longest dispersal distances (i.e., largest α values, Figure 4). It suggests that the study species are not limited to dispersal. The environmental components explained that the variation levels were quite small in all four metacommunities; the environmental predictor was more powerful in metacommunity A than in the other three metacommunities (Figure 5).

Locality connectivity was calculated at the largest α values (i.e., α parameter = 500 km) in the metacommunity A by connectivity metric C. Localities with high connectivity value across the whole study area, i.e., the stepping-stones, were found in the middle range of the riparian landscape, which was outside the protected area, and in the central part of mountain range, especially in the low mountains (Figure 1).

## 4. Discussions

High beta diversities with small components of nestedness were found in the mountain, riparian and desert landscapes of the study area. Bird species were not dispersal limited in these areas, and the bird species composition variations were explained mostly by locality connectivity rather than environment and spatial variation. The bird species showed higher environment references across the whole study area than in the homogenous landscape.

### 4.1. Bird Species Beta Diversity

The three landscapes in the study area have distinct vegetation profiles and topological structure characteristics. More than half of the bird species that were found in the mountain landscape were also recorded in the riparian and desert landscapes; thus, the plain area is also important for birds that live in the mountains and vice versa. The desert landscape has a higher beta diversity than that of the mountains and riparian landscapes, and its turnover component is also the highest, though it harbors the least number of bird species. This indicates that in the areas with poor vegetation conditions, there is also an obvious spatial variation of species, and the conservation of species distributed in special habitats may need to be expanded in the protection area.

The bird species compositions varied appreciably both within and across landscapes, reflecting very substantial beta diversities. Nestedness contributed much less than turnover; it indicated that the bird species found at less diverse sites were not subsets of species at more diverse sites. The bird diversity patterns suggest that not only the mountain landscape but also the riparian and desert landscapes, which are not in the range of the Altai-Sayan ecoregion, should be considered conservation hotspot areas for birds. Moreover, larger/more protected areas are needed to totally cover species turnover.

According to our research, the composition of birds in a particular area may change over time (Appendix C). As a result, long-term research should be taken into consideration to understand the dynamics of metacommunities, particularly in areas where the environment may change dramatically in response to climate change, and conservation strategies such as building artificial stepping-stones should be considered.

### 4.2. Contribution of Environment, Spatial Variation and Locality Connectivity to Bird Species Variation

The three sets of predictors, i.e., environment, space and locality connectivity, separately and significantly explained the variation in the regional community composition of bird species, though their contributions were different in the four metacommunities. When only one set of predictors was considered, bird species composition variation in metacommunities M, R and D were explained mostly in terms of connectivity, whereas for metacommunity A, the variations were largely due to space. Environmental factors were also explained more in metacommunity A than that in other monotonous landscapes. These indicated that space and environment influenced the bird species’ composition at a broader scale and in the landscape containing multiple habitats. 

The mechanism of species sorting through the differences in environmental affinities was more obvious in a heterogeneous landscape. Generally, environmental contribution was quite small compared with connectivity and spatial variation. As predicted by the Mantel corregrams of the environment spatial structure, there was environment autocorrelation within 100 km; thus, the contributions of the environment and space overlap within the spatial distance of 100 km. Beyond this distance, environment factors were not spatially autocorrelated and slightly influenced community composition across localities. The low environmental contribution suggested that the bird species have little affinity to the measured environmental features (i.e., non-specialist) within each landscape. Perhaps the environmental conditions within each landscape were not heterogeneous enough to separate the bird composition, given that they are not dispersal limited to suitable sites (see the next paragraph). 

Another explanation would be that not all of the important environmental features were measured; however, the spatial component is a potential proxy for missing spatialized environmental factors. In metacommunity R, the spatial predictors also explained very little variation, indicating that the important environmental features were considered; whereas in metacommunities A, M and D, the spatial predictors explained a certain amount of variation, indicating some important environmental features that might not be considered, like plant species composition. It is possible that the study missed important non-spatialized environmental predictors; for instance, foraging strength in the mountain landscape, farming effect in the riparian landscape, and the distance to the water source in the desert landscape. It is also possible that a particular species did not occupy suitable habitats in time for this study observation [28]. In addition, it is worth noting that HFP was an important environmental factor in all four metacommunities, and future studies should pay attention to the role of human activities in shaping metacommunity species composition.

For the connectivity predictor, the matrix based on the nearest occupied site does not vary with α, whereas the average connectivity matrix C does. It is the average connectivity metric C that largely contributed to explaining the amount of variation of bird species composition across localities with increasing α values in each metacommunity. The high predictive power of connectivity is in relation to the high levels of dispersal capacity, in which the average dispersal distance was estimated to not be limited to the longest distance between the localities. It suggests that the bird species are not limited in their capabilities of finding suitable habitats in the landscapes. Though low autocorrelation in the locality quality has been recognized as a challenge in terms of dispersal [29,30], it is not a challenge for birds in this study. In contrast to the connectivity predictor, there is significant theoretical and empirical evidence that environmental heterogeneity and spatial processes can have a large impact on metacommunities [17]. Connectivity is proved to contribute to explaining the single-species community variation in the metapopulation [24] and in a dispersal-limited soft-bottom, small marine metacommunity [15]. The importance of connectivity in terms of species persistence has also been theoretically proven by improving access to optimal resources [31,32].

## 5. Conclusions

This study demonstrated the impact of environment, space and locality connectivity on the high dispersal capacity of bird species in multiple landscapes. It may provide insight into the complex dynamics that determine species distribution. There are mechanisms that should be further considered in the metacommunity theory in the future. The measured grain size of the bird community and environmental variables would hinder the underlying process that driving metacommunity diversity [5]. The current understanding of metacommunities focuses on dynamics upon which spatial and local constraints occur at the same temporal scale. Extending the spatial scale to very large regions may involve processes that occur on historical timescales, leading to biogeographic rather than metacommunity dynamics [4,6]. Another crucial challenge is to predict the extinction rates of certain species and the functional consequences of an ecosystem due to their loss [31]. Furthermore, besides the spatial flows of species (dispersal), the spatial flows of energy and materials (resources) should also be quantified in a metacommunity at the ecosystem level [33].

## Figures and Tables

**Figure 1 animals-12-02341-f001:**
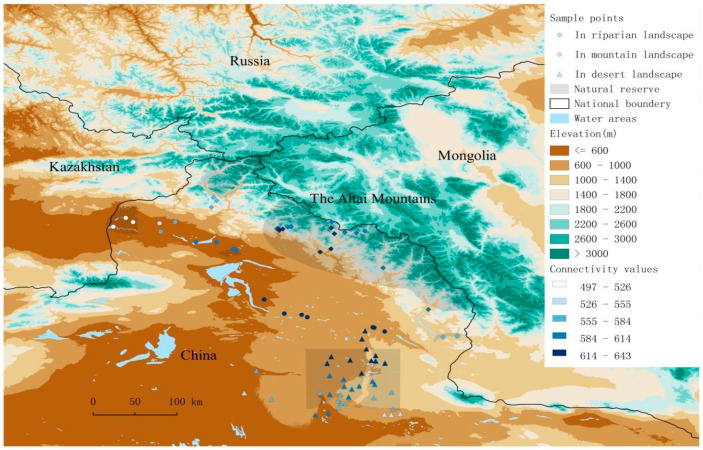
Study area and location of transect lines. Transect lines located in mountain landscape-square points; riparian landscape-circle points; desert landscape-triangular points. Connectivity values are showed as a graduated blue color from low value to high value.

**Figure 2 animals-12-02341-f002:**
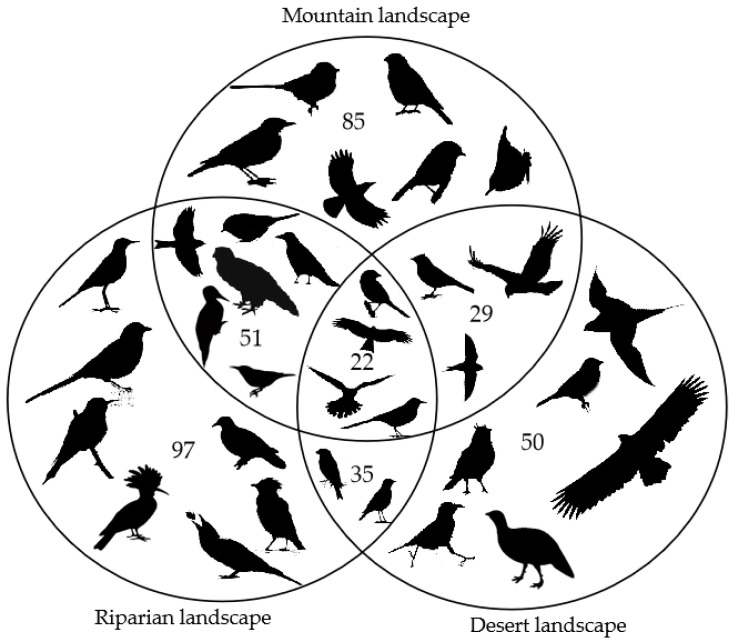
Number of bird species detected in the mountain, riparian and desert landscapes, shared by every pair of landscapes and by all three landscapes.

**Figure 3 animals-12-02341-f003:**
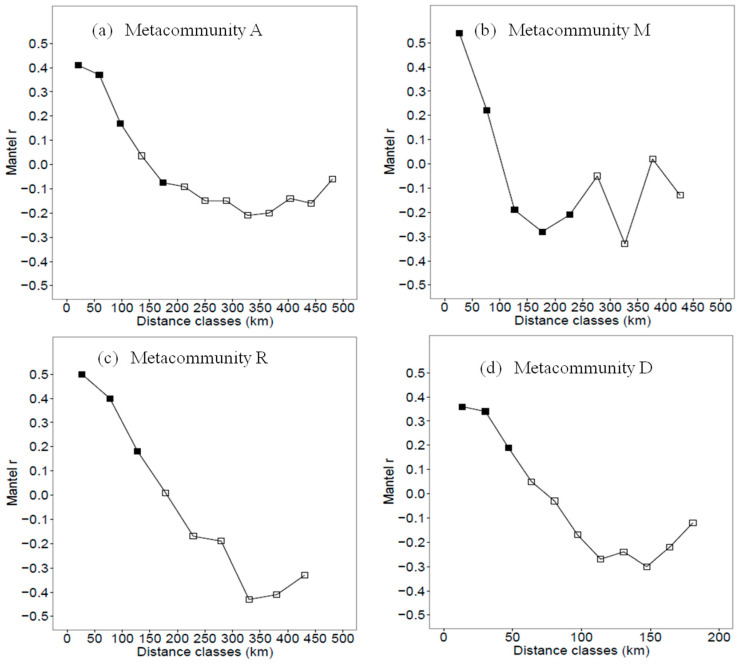
Mantel correlograms for environmental predictors in (**a**) metacommunity A (localities in the study area), (**b**) metacommunity M (localities in the mountain landscape), (**c**) metacommunity R (localities in the riparian landscape) and (**d**) metacommunity D (localities in the desert landscape). Solid squares indicate that the correlations are significant at a = 0.05, and the blank squares indicate that the correlations are not significant.

**Figure 4 animals-12-02341-f004:**
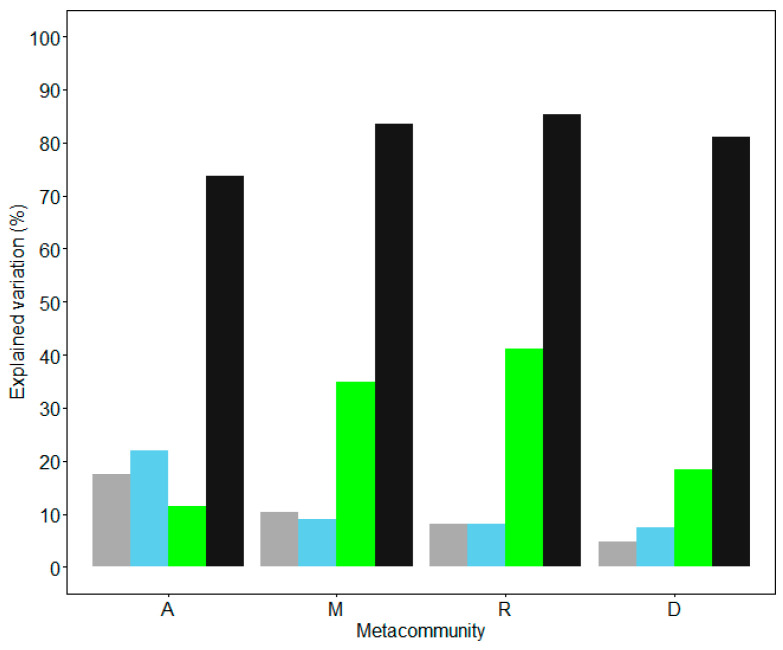
Variations of bird species composition in metacommunity A (localities in the whole study area), metacommunity M (localities in the mountain landscape), metacommunity R (localities in the riparian landscape) and metacommunity D (localities in the desert landscape) explained by only environment (grey), space (blue) or connectivity (green) predictor, and the maximum variation explained by all of the three predictors (black) at longest dispersal distances (i.e., largest α values). All of the models showed *p* < 0.01 by the Monte Carlo permutation test.

**Figure 5 animals-12-02341-f005:**
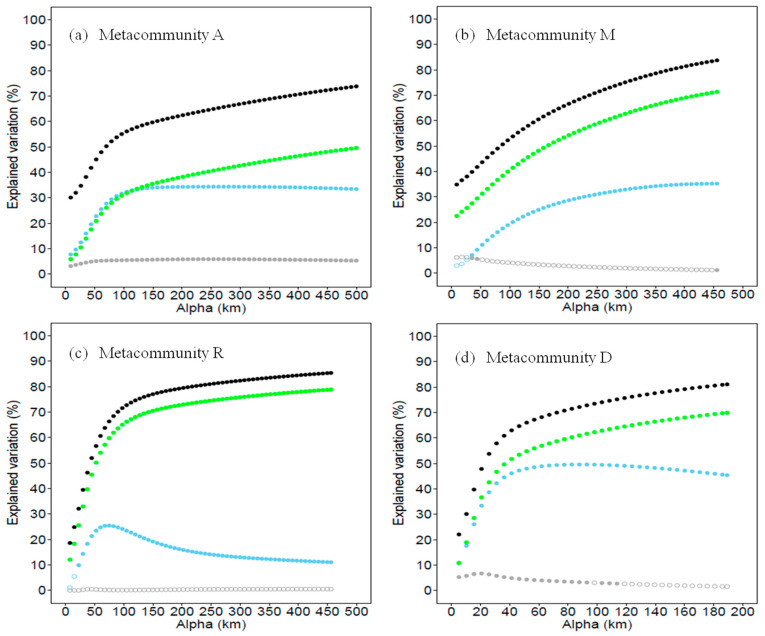
Changes in total explained variation (black point) and unique predictor explained variation as a function of α values in (**a**) metacommunity A (localities in the whole study area), (**b**) metacommunity M (localities in the mountain landscape), (**c**) metacommunity R (localities in the riparian landscape) and (**d**) metacommunity D (localities in the desert landscape). Due to the correlation with connectivity predictor (green points), the unique contributions of space (blue points) and environment (grey points) are also affected by α values. Solid points indicate the contribution is significant (*p* < 0.05, Monte Carlo permutest) and circles indicate contributions that are not significant.

**Table 1 animals-12-02341-t001:** Elevational range, number of transect lines (localities), number of species detected in more than two transect lines, pairwise dissimilarity values for Sørensen’s (beta.sor), Simpson’s (beta.sim), nestedness component (beta.nes), and the ratio of nestedness component to Sørensen’s dissimilarity (beta.ratio) in metacommunity A (localities in the whole study area), metacommunity M (localities in the mountain landscape), metacommunity R (localities in the riparian landscape) and metacommunity D (localities in the desert landscape). The Stand Error values of dissimilarity indices were calculated by locality pair compared.

Meta- Community	Elevation Range (m)	No. of Transect Lines	Species Richness	Beta.sor (±se)	Beta.sim (±se)	Beta.nes (±se)	Beta.ratio
A	443~2311	78	107	0.83 ± 0.004	0.73 ± 0.006	0.10 ± 0.004	11.6%
M	924~2311	22	71	0.63 ± 0.011	0.53 ± 0.022	0.11 ± 0.007	16.6%
R	414~1136	19	81	0.58 ± 0.010	0.42 ± 0.012	0.17 ± 0.011	28.7%
D	443~1329	37	48	0.72 ± 0.007	0.60 ± 0.010	0.12 ± 0.006	16.4%

## Data Availability

We are glad to archive all result-related datasets in any publicly accessible repository, such as Dryad, in appropriate forms as per requests as soon as this manuscript is accepted.

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
