# Peer review of "Bird Beta Diversity in Sharp Contrasting Altai Landscapes: Locality Connectivity Is the Influential Factor on Community Composition"

_animals, 2022, doi:10.3390/ani12182341_

Round 1

Reviewer 1 Report (Previous Reviewer 1)

I quite understand the rationale why the authors did not use recommended method. Al other issues seems to be corrected. I dont have any further comments

Author Response

We thank you so much for your comments and your patience, and thank you for helping us improve this manuscript.

Reviewer 2 Report (Previous Reviewer 3)

I thank the authors for correcting the manuscript and noting my suggestion of needed correction. I also thank them for any clarification on the questions posed. The resubmitted manuscript has been corrected by the Authors. My suggestions are reflected in the corrections made.

Only a few things are still to be corrected.

1.       I pointed out that ”The relationship analysis lacks a multivariate analytical approach that attempts to explain the relation between beta values and environmental factors”.

The authors responded “The relation between beta values and environmental factors are shown in Line 85-90: “In metacommunity A, all of the six variables significantly affected bird species composition. In metacommunity M, effective environmental variables were HFP, CTI, elevation, MDTR and AP. In metacommunity R, the environmental variables that significantly affected bird species composition were HFP, elevation, MDTR and AP. And in metacommunity D, effective environmental variables were HFP, elevation and MDTR.” And in Fig. 4.”

1.1. in lines 85-90 there is no such information about multivariate analysis;

1.2. perhaps the Authors meant lines 285-290, but the information in this paragraph "i. e. In metacommunity A, all of the six variables significantly affected bird species composition" is not clearly described and refers to the Mantel correlogram. The Mantel correlogram is the result of spatial correlation analysis, where on the X-axis we have a function of geographic distance classes between study sites (transects) and on the Y-axis the correlation coefficients. As I understand from the description of Mantel's methods, the correlation between the matrices was calculated, that is, between the multidimensional (multi-species) composition distance matrix of bird assemblages and the design matrix representing each class of geographic distance in turn. But there is no information on how much each variable affected the value of species exchange;

1.3. The authors in the paper evaluate three measures of beta, with the Sorensen coefficient being linearly related to beta diversity, but the Simpson index is not. The Simpson dissimilarity index (βsim) describes spatial turnover without the influence of species richness gradients. So wouldn't it be better to use a linear or non-linear analysis, defining as the dependent variable in one model the Sorensen beta value and in the second model the Simpson beta value. The main environmental variables can be used as independent in both models, and Species richness also in model 1. Such relationships are often logarithmic. Such a procedure would make it possible to ascertain the significance of the effect of each variable on the level of beta.

2.       “Birds were counted within 50 m on both sides of the transect (line 147) - this is a very short distance - does it follow from this description that observations above 50 m were omitted from the count? Since the distance of each individual encountered to the transect line was measured 2 (line 153-155) - such collected data could be processed according to the principles of Distance Sampling Methods and analysis of differential detectability due to species activity, and their behavior, etc”

The Authors responded. Response: We chose a distance of 50 meters because in woodland, birds within this distance can be detected 100% of the time. We made up for the missing information of birds from 50 m away with a longer transect length (5km).

2.1. In this case, it would be worthwhile to clarify it in the methods, that the description is only about the partial composition of the avifauna, it would be worthwhile to state what part of the data was rejected, and how many species of were not taken for analysis. The reader should know what percentage of the species composition is represented by the analyzed assemblages. This is a serious shortcoming of this work - after all, species composition is often evidenced by rare species inhabiting certain types of environments and quickly influenced by environmental changes.

3.       I also pointed out that conducted counts on transects are quite fast - 2 km/h, by different persons, and the aspects may play role in species detectability and the reliability of the compared data collected. I formulated questions - Did the different authors of the counts have the same species detection rate? Was this aspect tested? If so, it would be appropriate to provide the results of such testing of?

Authors Response: In this study area, based on preliminary experiment, 2km/h has a similar detectability with 1km/h ( not includes stop and record time). And the first author always here in the field research, the helpers are here to help record the detection on record chart and help to identify bird species. In fact among different years, bird species and abundance changed, that about ±5% in riparian and desert and ± 8% on mountains. So we only included the common bird species ( which detected in more than two transects) in analysis. We have deleted the sentence “To test whether bird species composition were consistent among different years, in every landscape, two transect lines were visited in every of the three years. “ in ms.

3.1. It would be worthwhile to provide this information in the paper - maybe show the results of attesting bird detection in the form of an Appendix.

It is worth including in the conclusion, that we are talking about only a part of the avifauna complex.

Author Response

This manuscript is a resubmission of an earlier submission. The following is a list of the peer review reports and author responses from that submission.

Round 1

Reviewer 1 Report

The ms deals with a comparison of bird communities among three main habitats within Altai Mountains. The study is well designed with sufficient number of localities that may uncover within habitat variability. The authors decided to compare beta diversity patterns, which seem to be suitable to show differences among habitats. Since these three habitats (riparian, desert and mountain) largely differ in vegetation structure, the authors found pronounced differences among bird communities. My major point is missing appropriate multivariate analysis that would clearly show the community structures within each habitat. Since authors use localities from large geographical scale, it would be useful to correct this analysis for geographical predictors. I recommend to use PCNM (Principal coordinates for neighbour matrices) analysis that is designed directly for these datasets (e.g. Legendre et al. 2008 ). So, primary predictors (i.e. response variables) will be abundances of each species (or their percentages within one locality), spatial predictors (PCO) will be then computed by the analysis based on coordinates. As environmental (i.e. independent variables) I suggest to use habitat type. This analysis may clearly show the differences between the habitats. The analysis can be performed using Canoco or R software. I suggest the results of this analysis can significantly improve the clarity of results. I have also following minor points:

L41, 97- Please check spelling of Altai.

L58 – Add space before the bracket.

L127 – Add space before “10”.

L137 – The expression “study set transect lines” seems not to be correct. Please, check it.

L155 – Add space before “The first”.

L168 – There is probably missing some word in the explanation of the formula. What is local slope in radians?

L186, 188 – Add space between “species” and “matrix”.

L187 – Add space between “columns” and “are”.

L192 – Replace “corrdinate” with “coordinate”.

L193 – Wrong spelling of “et al.”.

L195 – Replace “beta” with “Beta”.

L253 – Add space after the bracket.

L421 – Add author contributions.

L422 – Add description for Funding.

L423 – Add Institutional Reviw Board Statement.

L437 – The font in the Appendix legend seems to be too large.

References:

Legendre, P., Borcard, D and Peres-Neto, P. (2008) Analyzing or explaining beta diversity? Comment. Ecology 89, 3238–3244.

Reviewer 2 Report

This manuscript investigates metacommunity-level drivers of bird diversity across three landscape types (mountain, riparian and desert) and all three combined. Overall the work has been carried out well and is publishable quality. What detracts heavily from the manuscript currently is the quality of written English, which is not yet up to the required standard. As a result of this I found it hard to extract the main information content and would like to withhold my main review until a heavily edited re-submission.

My main feedback for the resubmission of this manuscript would be as follows:

1.      A full, hard edit throughout all sections of this manuscript, preferably by a native English speaker (or someone with high skills in written English). Two specific points to note are that I think the word ‘locality’ has been used throughout when ‘location’ is meant, and also ‘homogenous landscape’ is more meaningful than ‘monotonous landscape’. However, I have highlighted many (but by no means all) of the writing errors in the manuscript (uploaded).

2.      Much more detail is required in the methods section. For example, in section 2.2 was bird species detectability or distance decay from the transect line considered in any way? In section 2.3, EVI is a remote sensing derived measure. What data source was the EVI based on? Equally, what was the spatial resolution for the EVI, CTI, Human Footprint and WorldClim data? This matters in particular to understand the spatial resolution of the input variables against the 5km transect length.

3.      The Introduction seems rather short and generalised. I would expect a little more depth of consideration of the literature. Note that beta diversity has been studied since way before 2001, so the statement in lines 59-60 needs re-stating. Also, the final paragraph describes what the paper includes rather than states specific aims and objectives.

4.      The discussion section is also a little brief. Most significantly, a separate Conclusions section is required to make clear what the main outcomes of this work are.

Reviewer 3 Report

The paper assesses the composition of metacommunities of a large geographic area of the Altay region of China, covering three landscape types (riparian, desert, and mountain), and the analysis of the data and the results obtained may be relevant to understanding the basis of the rules shaping the distribution of species. The beta diversity is composed of turnover and nestedness processes, which were studied by the authors on a large sample of data. The environmental data used in the analysis are correctly selected, and sufficiently described, and the analysis performed taking into account the spatial autocorrelation of the data (Moran's coefficient) does not raise major objections. The redundancy analysis (RDA) and Moran's Eigenvector Maps analysis are well done. However, there are elements of the work, including methodological aspects, that would require better explanation, after all, data analysis and the results of this analysis depend primarily on what the input data are.

Introduction: while the introduction to the research topic presents the complex problem of metacommunities well, defines well the phenomena analyzed later in the teams, and is based on the new literature, quite against this background the research hypotheses are poorly, imprecisely formulated and the purpose of the research seems to be defined rather generally. 

Methods: 

1/ the pace of conducted counts on transects is quite fast - 2 km/h - Has it been tested what is the detectability of species under such methodological assumption? The authors wrote "The first author conducted bird counts, accompanied by two field assistants. Both the first author and field assistants were trained to identify local bird species in 2013 and 2014 before data collection. To test whether bird species composition among different years in these areas, two transect lines in every landscape were visited in each of the three years." It would be more interesting for readers and to assess the reliability of the compared data collected by the methods described in the paper's methodology to provide information resulting from the analysis- Did the different authors of the counts have the same species detection rate? - I have a question for the authors - was this aspect tested? If so, it would be appropriate to provide the results of such testing of species detectability? 

2/ Birds were counted within 50 m on both sides of the transect (line 147) - this is a very short distance - does it follow from this description that observations above 50 m were omitted from the count? Since the distance of each individual encountered to the transect line was measured (line 153-155) - such collected data could be processed according to the principles of Distance Sampling Methods and analysis of differential detectability due to species activity, and their behavior, etc.

3/ The authors include to species richness as a separate species Parus bokharensis. It was a separate species, but now it's a Parus major subspecies (Parus major bokharensis)  - this changes the number of species and results!

3/ The relationship analysis lacks a multivariate analytical approach that attempts to explain the relation between beta values and environmental factors.

Appendix S1

Latin nomenclature of many species is outdated. 

Latin species names should be reviewed and corrected (suggests seeing IOC World Bird List v12.1, Integrated Taxonomic Information System), some are not current nomenclature as a result of taxonomic, molecular studies, and genus name changes.

E.g. Azure tit Parus cyaneus is currently Cyanistes cyaneus 

Turkestan tit Parus bokharensis is currently Parus major bokharensis

Coal Tit Parus ater, now is Periparus ater

Picoides leucotos, it is Dendrocopos leucotos

Lesser Spotted Woodpecker Picoides minor now it is Dryobates minor

Willow Tit - is in the genus Poecile, not the genus Parus

Common Rosefinch is now the proper Latin name - Erythrina erythrina.

Lesser Whitethroat has been separated into the genus Curruca

Delichon urbica proper name Delichon urbicum

Sylvia nana is also the genus Curruca

Pallas's Leaf Warbler separated into a new genus, it is now no longer Phylloscopus, but Abrornis proregulus

Please correct also: Carduelis chloris, Parus palustris, Aquila clanga, Carduelis cannabina, Hippolais caligata, Sylvia communis

I don't feel qualified to judge about the English language and style, but I noticed grammatical errors in many places